# Effects of Temporary Respiration Exercise with Individual Harmonic Frequency on Blood Pressure and Autonomic Balance

**DOI:** 10.3390/ijerph192315676

**Published:** 2022-11-25

**Authors:** Sungchul Mun, Sangin Park, Sungyop Whang, Mincheol Whang

**Affiliations:** 1Department of Industrial Engineering, Jeonju Universtiy, Jeonju 55069, Republic of Korea; 2Industry-Academy Cooperation Team, Hanyang University, Seoul 04763, Republic of Korea; 3Rotary and Mission Systems, Lockheed Martin, 199 Borton Landing Road, Moorestown, NJ 08054, USA; 4Department of Human-Centered Artificial Intelligence, Sangmyung University, Seoul 03016, Republic of Korea

**Keywords:** human factors, autonomic balance, blood pressure, harmonic frequency, optimized respiration, heart rate variability

## Abstract

This study investigated the effects of modulated respiration on blood pressure and autonomic balance to develop a healthcare application system for stabilizing autonomic balance. Thirty-two participants were asked to perform self-regulated tasks with 18 different respiration sequences, and their electrocardiograms (ECG) and blood pressure were measured. Changes in cardiovascular system functions and blood pressure were compared between free-breathing and various respiration conditions. Systolic and diastolic blood pressures stabilized after individual harmonic breathing. Autonomic balance, characterized by heart rate variability, was also stabilized with brief respiration training according to harmonic frequency. Five machine-learning algorithms were used to classify the two opposing factors between the free and modulated breathing conditions. The random forest models outperformed the other classifiers in the training data of systolic blood pressure and heart rate variability. The mean areas under the curves (AUCs) were 0.82 for systolic blood pressure and 0.98 for heart rate variability. Our findings lend support that blood pressure and autonomic balance were improved by temporary harmonic frequency respiration. This study provides a self-regulated respiration system that can control and help stabilize blood pressure and autonomic balance, which would help reduce mental stress and enhance human task performance in various fields.

## 1. Introduction

In recent years, people have frequently been exposed to mentally stressful situations at their workplaces, in which cognitive mechanisms in the human brain simultaneously process numerous pieces of information. It is very important to control mental workload or stress while performing work or daily activities under a variety of circumstances [1,2,3,4,5,6,7,8,9,10]. They often refer healthcare professionals to ask for effective ways to manage the level of mental stress themselves or wish to use health monitoring systems before serious diseases occur [11,12,13]. Among those recommended, a controlled respiration method, generally known as deep breathing, has often been suggested to modulate physiological mechanisms underlying mental stress and homeostasis. However, how people control respiration has rarely been studied. People are exposed to different neurophysiological and environmental factors, and individual optimal frequencies and times regarding self-respiration training are yet to be determined.

Many studies have found that controlling respiration alone can reduce the symptoms of major disorders [14], enhance the efficiency of pulmonary gas exchange [15], improve symptoms of fibromyalgia [16], and balance the relationship between sympathetic and parasympathetic regulation [17]. Such findings have supplemented various ways of improving people’s health regarding blood pressure diseases and autonomic balance imperfections, but few studies have focused on how respiratory processes can affect and supplement people’s health, especially with blood pressure and autonomic balance. Currently, blood pressure and autonomic balance are considered clinical markers of human health due to the consequences of their irregularities. Unbalanced blood pressure can result in neurological damage, such as clinical shock and, in a worst-case scenario, death [18,19,20]. In addition, irregular autonomic balance, which describes the balance between the functions of the sympathetic and parasympathetic branches, could result in heart attacks and, in the case of blood pressure, death [21]. Thus, it is important to investigate how respiratory processes modulate blood pressure and autonomic balance.

Autonomic balance and respiration have been found to be closely correlated [22]. These are controlled by the vagus nerve. This nerve partially controls the parasympathetic innervation of autonomic balance, which is affected by respiration through the thoracic ganglia [23]. Inhalation obstructs vagal activity, causing the heart rate to increase, whereas exhalation allows vagal activity to resume, causing the heart rate to slow [23,24]. Variations in the heart rate caused by vagal activity is known as heart rate variability (HRV). The vagal activities of autonomic balance, also known as HRV, and respiration are defined as respiratory sinus arrhythmia (RSA) [25,26,27]. HRV, or variation in the parasympathetic and sympathetic responses, is also partially controlled by the baroreflex system. It is known that blood pressure is affected by cardiac output or regulated by baroreceptors. Many studies have found that HRV and blood pressure are related through the baroreflex system [25,26]. Figure 1 shows a summary of the relationship between respiration, autonomic balance or heart rate, and blood pressure.

Given this relationship, a control method can be developed whereby controlling respiration and autonomic balance could conclusively manipulate blood pressure. The primary purpose of the control is to manipulate the respiration frequency and, by doing so, manipulate the autonomic balance frequency to the same or similar frequency. With two identical frequencies, the amplitude doubles as the frequencies construct with each other. A few papers already have studied and found the optimal frequency, which is known as the “Natural Harmonic Frequency,” of the arterial baroreflex that initiates this phenomenon and its effects [28]. The natural harmonic frequency is the original frequency of the cardiovascular system (CVS). If the CVS is matched to this frequency, all physiological function frequencies will be manipulated to the same frequency; thus, these frequencies will merge together, and the functions will stabilize to their optimal range. The average natural harmonic frequency was found to be 0.1 hertz (Hz), which varies slightly from person to person by approximately ±0.5 Hz [27]. Controlling the breathing frequency to the appropriate natural frequency should stabilize the frequencies of HRV and blood pressure and eventually harmonize the frequencies and amplify their effects on the body. Thus, this study aimed to investigate the effects of individual harmonic frequencies on autonomic balance and blood pressure stabilization. This study mainly investigated and observed the effect of respiratory control on blood pressure and autonomic balance, and developed a reasonable method for mobile healthcare applications.

## 2. Methods

### 2.1. Participants

Written informed consent was obtained from each participant prior to the experiment. The Institutional Review Board at Sangmyung University in Seoul, Republic of Korea approved all the protocols used in this study. Thirty-two participants, sixteen women and sixteen men, with ages ranging from 23 to 32 years (mean: 26.43 years), voluntarily participated in the study; incentives of USD 40 were offered to encourage their participation. Only participants with no history of any disorders, particularly cardiovascular diseases that could have affected their heart rates, were included in this experiment. One day prior to the experiment, to minimize possible variations in cardiovascular responses, participants were prohibited from smoking and consuming alcohol or caffeine. Additionally, they were recommended to have a full night of sleep. Before the experiment, participants were informed of the experimental protocols and procedures, except its purpose.

### 2.2. Experimental Protocol

The room designed for the experiment included a chair and a screen placed in front of the chair. The participants sat on a chair during all trials of the experiment. Before the respiratory experiment, each participant was given a 5 min break to eliminate external influences on both autonomic balance and blood pressure [29,30]. After the rest period, each participant’s blood pressure was recorded using an armband. When a participant’s blood pressure reached the recommended range (systolic pressure: 90–119 mmHg; diastolic pressure: 60–79 mmHg; autonomic balance, reference zone), the experiment commenced [31,32].

As mentioned before, previous studies have proved that the natural harmonic frequency averages at approximately 0.1 Hz [33]. However, studies have also mentioned that the frequency varies slightly from person to person. Natural frequencies are slightly different for each individual; therefore, steps were required to determine the harmonic frequency of each participant. These steps were based on previous studies that focused on determining an individual’s harmonic frequency [25,33]. The 0.1 Hz of the respiration frequency constituted of six cycles per minute. Previous studies have found that the range of people’s harmonic frequency of respiration is 4 to 12 breathing cycles per minute [25,28]. Therefore, each breathing cycle totaled 10 breathing cycles, including the free-respiration cycle. The heart rate of each participant was measured using electrocardiograms (ECG). According to previous findings, the individual harmonic frequency was determined by observing the peak amplitudes of the LF band of HRV for all breathing cycles. The breathing cycles that resulted in the highest peak in HRV were determined to be harmonic frequencies.

To help participants easily follow each breathing pattern, a video of a pump was played on the screen; the pump gage was filled and emptied according to the corresponding breathing cycle for inhalation and exhalation processes. Each participant’s blood pressure was measured before the breathing cycle; for comparison, blood pressure was also measured after the breathing cycle. Respiration (RSP) was measured with an RSP sensor, a band that goes around the participant’s chest, to observe and check if they were correctly following the breathing pattern. After each breathing cycle, a short break was provided to normalize the participants’ physiological signals. An illustration of the experimental procedure is shown in Figure 2.

### 2.3. Data Collection and Signal Processing

ECG signals were recorded at a 500 Hz sampling rate from Lead-I type, and the R-peaks were detected using QRS detection algorithms given by Pan and Tompkins [34]. The R-peaks were located by following these five steps [31,32]: (1) ECG signals, to attenuate noise, were passed through a band-pass filter (5–11 Hz); (2) the signal was differentiated to clear the inflection point of the QRS wave; (3) the signal was amplified by squaring; (4) to obtain information about the R wave, the moving average technique was used (window size: 150 ms; resolution: 2 ms); and (5) the R-peak was detected based on a threshold. This threshold was defined as the difference between the maximum and minimum values of the filtered ECG signal. The R-peak to R-peak interval (RRI) was calculated using the difference between R-peak to R-peak, and detected RRIs were filtered as normal to normal (NN) intervals of 600–1200 ms, based on previous research [35]. Time domain indices of cardiac activity such as standard deviation of normal-to-normal intervals (SDNN), root-mean-square of successive differences (rMSSD), and proportion of NN50 (pNN50) were extracted as follows. (1) The SDNN was computed as the standard deviation across normal RRIs. (2) The rMSSD was calculated as the root mean square of successive differences between normal heartbeats. (3) The pNN50 was calculated by the percentage of adjacent RR intervals that are >50 ms.

Successive RRI values in normal range were converted to a time series by resampling at 2 Hz. The resampled data were processed using a fast Fourier transformation (FFT). HRV spectra (0–0.4 Hz) were then obtained. The HRV spectrum was used to identify individual harmonic respiration frequencies and autonomic balance. An overview of the signal processing is shown in Figure 3. All signal processing and analysis on HRV parameters were conducted using LabVIEW 2010 software (National Instruments, Austin, TX, USA). The Welch’s method was used to calculate power spectrum density in the frequency domain parameters of cardiac activity.

An individual’s harmonic respiration frequency was defined as the highest amplitude value of the LF band (0.04–0.15 Hz) in the HRV spectrum band (0–0.4 Hz). All respiration tasks (4–12 breaths per minute with a 0.5 min interval and a total of 17 respiratory conditions), excluding the free respiration condition, were extracted from the LF band. The dominant amplitude in the LF band was then compared with that in the other respiratory conditions. The highest amplitude value among the respiratory conditions was determined as the individual’s harmonic respiration frequency, as shown in Figure 4, which is an example of harmonic respiration in participant 1 based on the highest amplitude value among the respiratory conditions.

The normalized HRV technique evaluates autonomic balance based on the sympathetic and parasympathetic nervous systems. The HRV spectrum was categorized into two frequency bands: the LF (0.04–0.15 Hz) and HF (0.15–0.4 Hz) bands. The power spectrum values (LF and HF) were converted into natural logarithms (assuming ln), and nine zones (zones 1–8 plus a reference zone) were created to categorize the autonomic balance (i.e., sympathovagal balance) [21,34,36], as shown in Figure 5. The zone descriptions for autonomic balance are as follows.
Zone 1: High parasympathetic/low sympathetic.Zone 2: High parasympathetic/normal sympathetic.Zone 3: High dual autonomic tone.Zone 4: High sympathetic/normal parasympathetic.Zone 5: High sympathetic/low parasympathetic.Zone 6: Normal sympathetic/low parasympathetic.Zone 7: Low sympathetic and parasympathetic.Zone 8: Low sympathetic/normal parasympathetic.

RSP was measured to identify breaths per minute. During the respiration task, RSP signals were recorded at a 500 Hz sampling rate from the thoracic region. This signal was processed using the FFT, and the frequency of the dominant peak was obtained. These values were then compared for each respiratory condition. It was also used to set a standard for task performance to determine the task performance accuracy of each participant. If a subject could not concentrate on the respiration task where the respiratory data did not match the controlled respiratory rates, such data were excluded from this experiment.

The ECG sensor used in the experiment was an SS29L multi-lead ECG cable (BIOPAC System Inc., Goleta, CA, USA) and was operated using Lead-I. The RSP sensor was a TSD201 respiration transducer (BIOPAC System Inc.), which measured the thoracic expansion and contraction. The ECG and RSP signals were recorded using an ECG 100C and RSP 100C amplifier system (BIOPAC System Inc.) and digitalized using NI-DAQ-Pad9205 (BIOPAC System Inc.) at a rate of 500 Hz. Signals were processed using an MP150 data acquisition unit, the AcKnowledge analysis software v 4.1 (BIOPAC System Inc.), and the LabVIEW 2010 software (National Instruments, Austin, TX, USA). An HEM-780 (OMRON system Inc., Kyoto, Japan) was used to measure blood pressure.

### 2.4. Data Analysis

The individual harmonic frequency of each participant was determined by comparing the HRV spectra (the highest amplitude value in the LF band) between the respiration tasks. To identify the effect of the breathing rate, the optimized breathing rate (individual harmonic frequency) was compared with other breathing conditions, including free respiration. Variations in autonomic balance and blood pressure caused by the breathing rate condition were measured. All the data were recorded before and after the respiration task. However, because the optimized breathing rate differed among participants, the personalized harmonic frequency breathing rate was compared with free breathing using SPSS 17.0 (SPSS IBM, Inc., Chicago, IL, USA). Autonomic balance was evaluated based on nine domains (zones 1 to 8, and the reference zone). Systolic and diastolic blood pressures were measured. To compensate for multiple comparisons, Bonferroni correction was performed for the derived statistical significances [34,37,38]. Statistical significance levels were controlled based on the number of independent hypotheses (i.e., *α* = 0.05/n); thus, *α* = 0.007 (blood pressure, autonomic balance, and time domain index, *α* = 0.05/7) was used. The effect size, based on Cohen’s *d* (parametric), was calculated to confirm the practical significance of the findings. For Cohen’s *d*, standard values of 0.20, 0.50, and 0.80 for the effect size are generally regarded as small, medium, and large effects, respectively [39].

The required sample size for our experiment design was determined using G*Power software v.3.1.9.7 (Heinrich Heine University Düsseldorf, Düsseldorf, Germany) with the following parameters: (1) tails = two tailed test. (2) effect size (Cohen’s *d*) = 0.8 (large effect size). (3) *α* = 0.01. (4) 1 − *β* = 0.95. The recommended sample size was obtained at 32 from G*Power software v.3.1.9.7, and a total sample size in this experiment (i.e., 64 samples) was tenable to achieve acceptable statistical power.

## 3. Results

Each participant underwent breathing cycles and blood pressure was measured. Using these data, the harmonic frequency was determined through an HRV spectrum analysis. Meanwhile, the breathing effects of the harmonic frequency and other breathing cycles on blood pressure and autonomic balance were compared. In conclusion, the effect of harmonic frequency of respiration on an individual’s blood pressure and autonomic balance was confirmed.

### 3.1. Harmonic Frequency

Table 1 presents the harmonic frequencies of the participants. As mentioned above, the breathing frequency which showed the highest HRV spectrum value was determined to be the harmonic frequency of an individual. For participant 1, in Figure 4, the highest HRV spectrum value was achieved when the participant was breathing 5.5 times per minute, and thus, the breathing cycle was deemed as the individual’s harmonic frequency. The harmonic frequency varied across participants and ranged from 4 breaths/min to 12 breaths/min.

### 3.2. Blood Pressure

The blood pressure during the harmonic frequency was compared with the blood pressure during other breathing cycles. During the harmonic breathing frequency, both the systolic and diastolic blood pressures of the participants were stabilized, and the blood pressures were increased or decreased to the optimal range. Meanwhile, during other breathing cycles, the systolic, diastolic, or both blood pressures failed to reside in the optimal range. In some cases, the blood pressure of a few participants reached the optimal range in multiple breathing cycles. However, these breathing cycles were similar to their harmonic breathing cycles. For example, participant 8 had a harmonic frequency of 8 breaths/min, and the participant’s blood pressure reached the optimal range at 7.5 and 8 breaths/min.

All participants had different harmonic frequencies; therefore, the free-breathing condition was used as a control for comparison. The effects of the harmonic frequency on the free-breathing cycle were compared for all participants. While the participants’ blood pressures were not within the optimal range during free respiration, the blood pressure was stabilized to the optimal range during the harmonic breathing cycle. As shown in Figure 6 and Figure 7, blood pressure was stabilized to the optimal range regardless of whether it was lower or higher than the optimal range. If the blood pressure was high, it decreased to the optimal range, and vice versa. Normality tests were assumed (*p* > 0.05) and paired *t*-tests were conducted. For the resonance frequency condition, systolic blood pressure in the post-task condition was significantly reduced compared with that in the pre-task condition (*t*(31) = 4.438, *p* < 0.001, Cohen’s *d* = 0.785, indicating a medium effect size) as shown in Figure 6. Significantly reduced diastolic blood pressures were also observed (*t*(31) = 3.679, *p* = 0.001, Cohen’s *d* = 0.650, indicating a medium effect size) as shown in Figure 7. In the free respiration condition, no significant changes were observed in systolic blood pressure (*t*(31) = 0.650, *p* = 0.521) or diastolic blood pressure (*t*(31) = 0.906, *p* = 0.372) of the participants.

### 3.3. Autonomic Balance

Breathing at the harmonic frequency also stabilized the autonomic balance of the participants, as the values of autonomic balance fell within the reference zone while other respiratory cycles failed to do so. As shown in Figure 8, the autonomic balance of participant 3 resided within the reference zone only during the harmonic frequency, whereas it resided in zones 2, 4, 5, 6, and 8 during other breathing conditions. For participant 8, autonomic balance was also stabilized while residing in the recommended zone for 7.5, 8, and 8.5 breathing cycles including the harmonic frequency. This is illustrated in Figure 9. Similarly to the case with blood pressure, the optimal breathing cycles of the participants were around the harmonic breathing cycle.

During free-breathing cycles, nearly all participants’ autonomic balances were in zones 4 and 5. These zones describe the sympathetic nervous system as highly active and parasympathetic, with little or no activity. Some participants also had autonomic balances residing in zones 2, 3, 6, and 8, but none were in the reference zone during free respiration, as shown in Figure 10.

The normality assumption was tenable (*p* > 0.05) and paired *t*-tests were conducted. Paired *t*-tests indicated that ln LF in the resonance frequency condition significantly decreased (*t*(31) = 10.814, *p* < 0.001, Cohen’s *d* = 1.912, indicating a high effect size) and ln HF power significantly increased relative to those of the free respiration condition (*t*(31) = 7.046, *p* < 0.001, Cohen’s *d* = 1.246, indicating a high effect size) as shown in Figure 10.

### 3.4. Time Domain Parameters of Cardiac Activity

As for the time domain data of cardiac activity, normality assumption was tenable (*p* > 0.05) and paired *t*-tests were conducted. The paired *t*-tests showed that the time domain parameters of cardiac activity in the resonance frequency condition were significantly higher than those in the free respiration condition: SDNN (*t*[31] = −4.154, *p* < 0.001, Cohen’s *d* = 0.780), rMSSD (*t*[31] = −3.113, *p* = 0.004, Cohen’s *d* = 0.700), and pNN50 (*t*[31] = −3.445, *p* = 0.001, Cohen’s *d* = 0.645). The mean (M) and standard deviation (SD) values were as follows: SDNN (Free: M = 79.874, SD = 64.607; Resonance: M = 99.671, SD = 25.330), rMSSD (Free: M = 27.573, SD = 1.169; Resonance: M = 28.737, SD = 2.040), and pNN50 (Free: M = 10.723, SD = 5.526; Resonance: M = 14.191, SD = 5.226), as shown in Figure 11.

### 3.5. Classification between the Free and Harmonic Respiration Conditions

The significant biomarkers demonstrated in this study (i.e., blood pressure and autonomic balance) were classified using machine learning algorithms to develop a mobile healthcare application for harmonic respiration. Before the biomarkers were trained and classified, resampling-based permutation tests were performed to further verify that significant differences would not be observed by chance. One-thousand permutations of the group variables were randomly generated for each blood pressure and heart rate variability indicator. As shown in Figure 12, the computed permutation *p*-values were in the adjusted critical region (i.e., extreme side of the permuted distributions), implying that no significant differences were found by chance. Five machine learning algorithms (logistic regression (LR), linear discriminant analysis (LDA), decision tree (DT), random forest (RF), and linear kernel support vector machine (LSVM)) were used to classify the two opposite factors of the biomarkers. The significant features were trained using the algorithms on 32 participant datasets with five-fold cross-validation. Classification performances were compared based on the receiver operating characteristic (ROC) curve and area under the curve (AUC) derived from confusion matrix indices (i.e., accuracy, sensitivity, and specificity). Python 3.9.2 (2021, Python Software Foundation) was used to conduct the classification and cross-validation. As for the Python code used, see Appendix A. As shown in Figure 13 and Figure 14, the RF models outperformed the other classifiers for the training datasets of systolic blood pressure and heart rate variability. The mean AUCs of the RFs were 0.82 for the systolic blood pressure and 0.98 for the heart rate variability. However, for the diastolic blood pressure of the training datasets and the three biomarkers of the test datasets, other algorithms (for example, LR, LDA, and LSVM) showed better performance in classification than the RF models, because of the limitation of the single feature and the small number of datasets for the test datasets.

A respiration training system with individual harmonic frequencies was developed using Visual C++ 2013 (Microsoft, Redmond, Washington, United States) and OpenCV 3.0.0 (OpenCV, Palo Alto, CA, USA), as shown in Figure 15. During the offline training stage, users are asked to perform slow breathing training according to the proposed slow breathing frequencies while their blood pressure and ECG data are recorded through wearable sensors. Resampling-based permutation tests validate whether reliable significant changes are observed among the breathing conditions. Then, an individual cardiovascular harmonic frequency was determined, and only significant parameters are trained and classified. The system automatically selects the best-performing machine learning algorithm. During the online testing stage, users are requested to only carry out two free and slow breathing as a function of the determined harmonic frequency for about 7 min with an interval of 1 min. The classifier selected from the best-performing model automatically categorizes significant parameters from blood pressure and ECG data into a binary class between the free and individual harmonic breathing. Then, classification results obtained from the offline and online stage are compared. Unless a clear classification performance is not reported, or if less than mean AUC of 0.70 is reported, the classifier is initialized, and a fine-tuning stage of individualization begins to optimize the parameters with different weights. The mobile healthcare application could determine the individual resonance frequency, which contributes to stabilizing the autonomic balance based on the classification procedures.

## 4. Discussion

The primary purpose of this study was to investigate the effect of individual harmonic breathing frequencies on blood pressure and the autonomic nervous system. The harmonic frequencies for each participant were determined using the highest LF power among the 17 respiratory conditions, and significant effects were observed. The study clearly showed that temporarily controlled respiration positively influenced blood pressure and the autonomic nervous system, controlling respiration to match the harmonic-frequency-stabilized blood pressure, whether it was high or low, to the optimal range, and the autonomic nervous system to the reference zone. Overall, our findings can be summarized into two significant statements: (1) blood pressure was stabilized to the optimal range, and (2) autonomic balance was stabilized to the reference zone with brief harmonic frequency respiration training.

While this study focused on the effects of harmonic frequency breathing on blood pressure and autonomic balance, several studies have investigated how respiration affects other functions from a physiological or psychological perspective [40,41,42,43,44,45,46]. One study showed that controlling breathing cycles to the harmonic breathing frequency of eight cycles per minute caused a decrease in anxiety levels compared with free respiration. Furthermore, the study found that other breathing frequencies did not show any stabilizing effects on anxiety or stress levels [47,48]. Other studies showed that “shallow” breathing patterns increased anxiety levels more than patterned breathing. Although these studies focused on stress and anxiety, they showed that controlled breathing has many benefits, further emphasizing the significance of harmonic-frequency breathing. Moreover, the act of lowering stress levels is highly correlated with blood pressure stabilization [49]. Another study resulted in the phenomenon, where controlling respiratory rates and thoracic breathing resulted in a drop in arterial CO_2_ levels [50]. These studies corroborate the findings of our study. Merely controlling breathing patterns to the harmonic frequency allows the blood pressure to fall or rise to the optimal range. In addition, for some people, breathing close to the harmonic frequency stabilized their blood pressure. Although, we have mainly focused on potential effects of short-term slow breathing training based on individual harmonic frequencies on autonomic balance, there might have been potential for extending the controlled harmonic breathing rates from the current slow breathing to normal breathing rates. This is another research topic that needs further experiments and studies to investigate physiological mechanisms underlying the harmonic respiration.

This study found that breathing to harmonic frequencies not only stabilizes the blood pressure, but also the autonomic nervous system. This was due to alterations in the volumetric flow of the heart, which were induced by modifications in respiration. A previous study showed that depending on the respiratory effort, cardiac output was altered by changes in heart rate and/or stroke volume [51]. Therefore, stabilizing the breathing pattern balanced the stroke volume, leading to cardiac output stabilization. Other studies also found that respiration was a powerful source of vagal afferent inputs to the central nervous system. Specifically, a reflex mechanism caused the heart rate to accelerate and inhibited the neural constrictor to other peripheral vascular beds as the lungs inflated [52]. Similarly, with the values of blood pressure, this study found that simply controlling the breathing frequency to the harmonic frequency not only allowed the participant to control the autonomic nervous system but also to reside within the reference zone. However, our study has the following limitations that need further experiments. Although autonomic balance was stabilized after the temporary harmonic breathing as a function of each individual resonance frequency, long-term effects of the breathing protocol remain to be seen with the further studies. We encourage researchers to investigate the effects of the harmonic breathing on stabilization in autonomic balance with a properly designed experimental protocol for the long-term effects. On top of that, the autonomic balance may be influenced by other potential factors (e.g., changes in peripheral vascular resistance, altered airway caliber, and changes in blood flow to other viscera). Notably, potential effects of cardiac vagal activity and vagal nerve mechanisms emphasized in regulating cardiac functioning, which are correlated with the autonomic balance, on sympathetic and parasympathetic regulation, are too complex to be understood and have yet to be determined. Thus, future studies are needed to determine that the other factors may be involved in the autonomic balance control with larger number of participants, varied time, repetitions of the respiration training characterized by each individual harmonic frequency, and other measurements of neurotransmitter levels.

Our study suggested the easiest way to stabilize autonomic balance and blood pressure through temporary harmonic respiration. Harmonic frequency can be defined as the frequency of the arterial baroreflex system [25,28,33]. By controlling respiration to the “natural” frequency of the arterial baroreflex system, the blood pressure and autonomic nervous system were controlled and balanced. This was observed in a study comparing respiratory effects on blood pressure and the autonomic nervous system between free respiration and breathing with harmonic frequency. Furthermore, the graph of systolic and diastolic blood pressures clearly illustrated the effects of breathing harmonic frequency on the human body. Significantly, the blood pressures for the participants were not stable before the experiment, but stabilized after the breathing experiments. In addition, for a few participants, the blood pressure and autonomic nervous system were stabilized not only at the harmonic frequency, but also at frequencies close to it. Hence, this study proved that people have unique harmonic frequencies that are not limited to a single value. Our data lend support to the idea that people have unique harmonic frequencies that are not limited to a single value, and breathing synchrony with the cardiovascular harmonic frequency can improve overall sympathetic and parasympathetic balance. Our findings suggest that modulation of respiration as a function of individual harmonic frequency can lead to reduced mental stress, by stabilizing blood pressure and the autonomic nervous system, because cardiac functions are closely tied to helping manage mental stress. It is expected that people can easily control their cardiac and mental health status if practical mobile applications based on wearable and comfortable ECG sensors are further developed.

## 5. Conclusions

This study found that matching breathing rates to an individual’s harmonic frequency had a positive effect on the body by balancing blood pressure and autonomic balance. These significant findings provide evidence that blood pressure and autonomic balance can be continuously stabilized solely through harmonic frequency breathing. Respiration is a voluntary process that controls other involuntary functions. Moreover, this study focused only on people with no pre-existing health conditions. Consequently, it cannot be concluded that the method of breathing with harmonic frequency can alleviate hypertension or cardiovascular diseases, but it may contribute to the prevention of these disorders. Therefore, future studies should be directed towards determining the effects of harmonic frequency breathing in patients with hypertension and other cardiovascular diseases.

## Figures and Tables

**Figure 1 ijerph-19-15676-f001:**
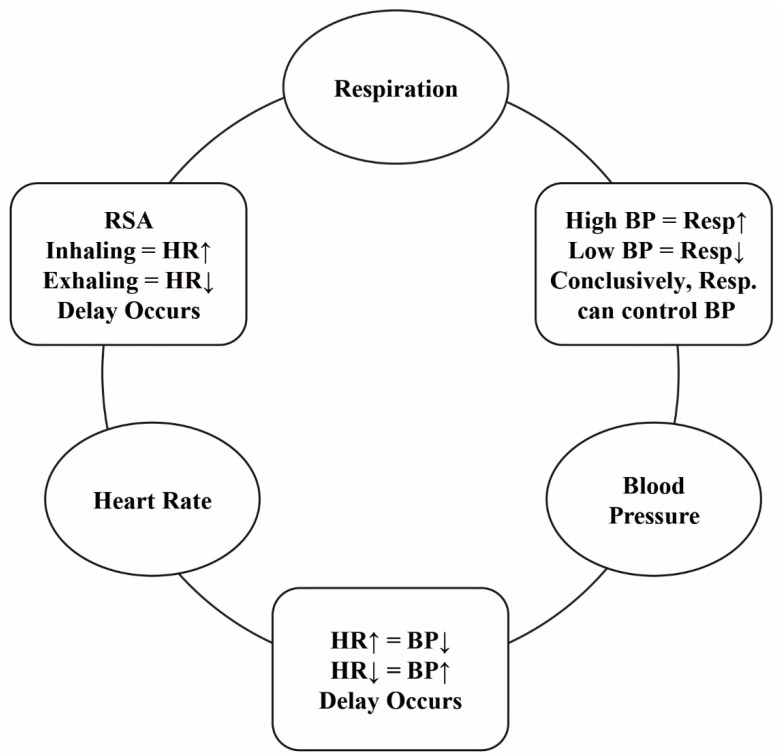
Overview of the closed system between heart rate variability, blood pressure, and respiration.

**Figure 2 ijerph-19-15676-f002:**
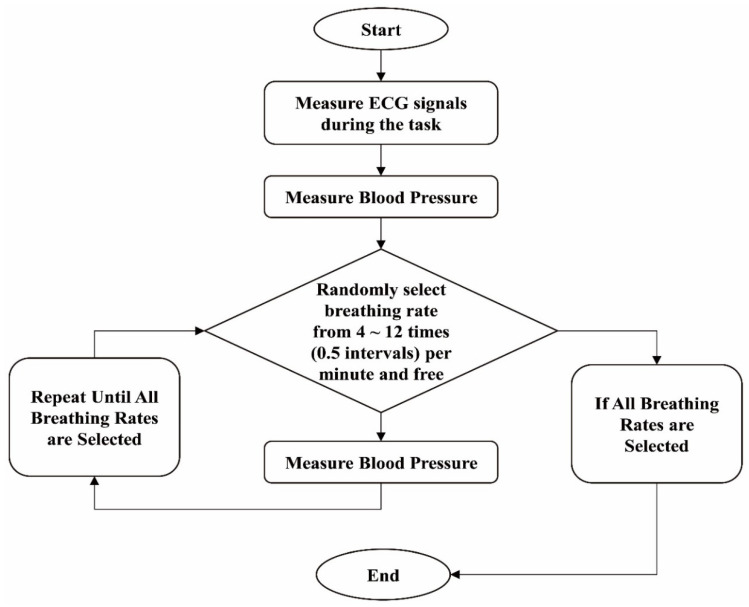
The experimental process.

**Figure 3 ijerph-19-15676-f003:**
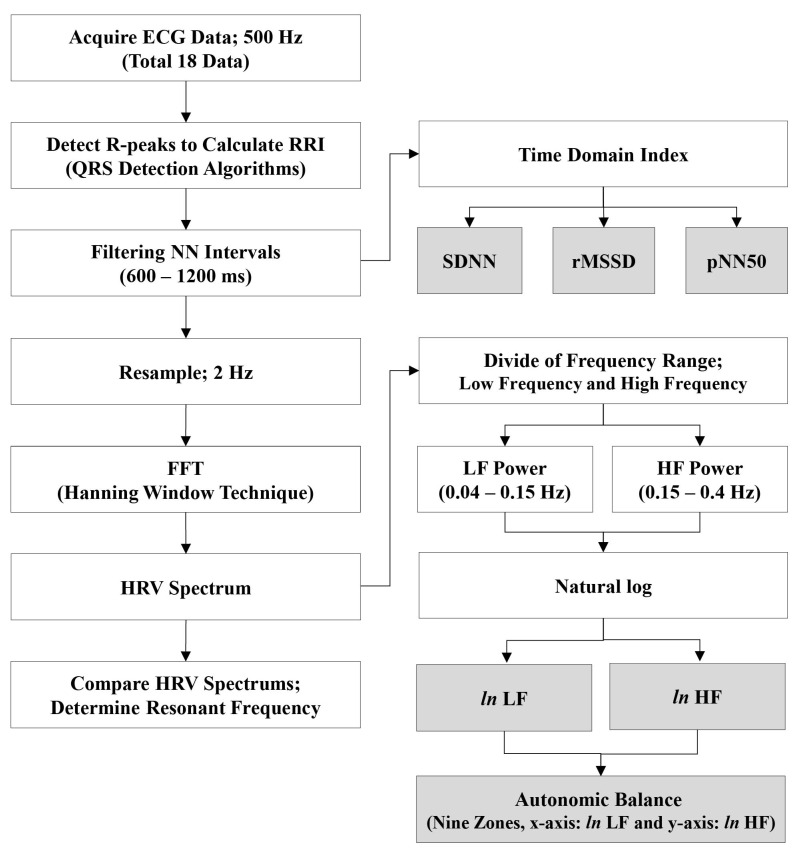
Overview of the signal processing to achieve resonant frequency. Top gray boxes represent the time domain index of the ECG data and the bottom gray boxes represent natural log values of LF and HF characterized for autonomic balance with nine zones.

**Figure 4 ijerph-19-15676-f004:**
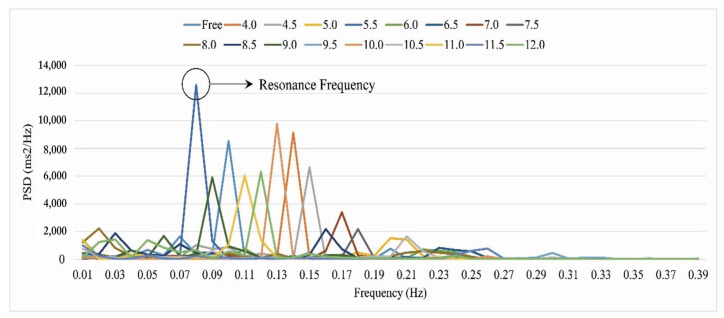
Example of harmonic frequency in participant 1 based on the highest amplitude value among the respiration conditions.

**Figure 5 ijerph-19-15676-f005:**
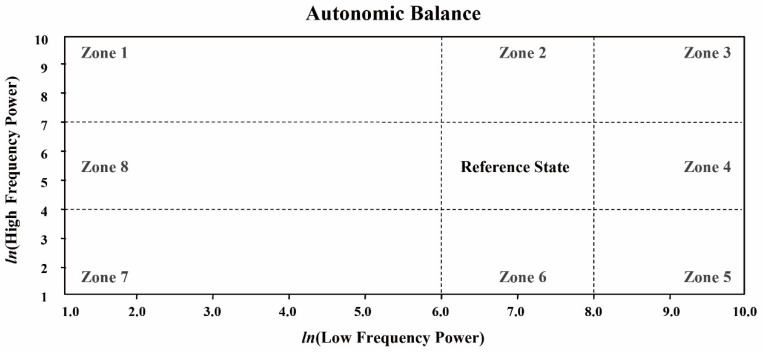
The zone of autonomic balance using HRV spectrum analysis.

**Figure 6 ijerph-19-15676-f006:**
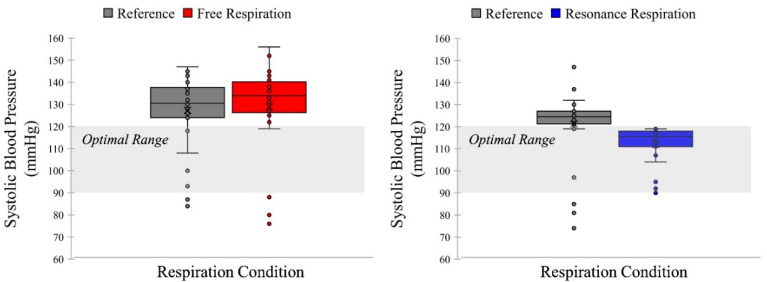
Comparison of results of systolic blood pressure between free respiration and resonance frequency. The resonance frequency was within the optimal range after respiration task, unrelated to the state before the task. On the other hand, the free respiration condition was within the (pre)hypertension state; thus, it was not in the optimal range.

**Figure 7 ijerph-19-15676-f007:**
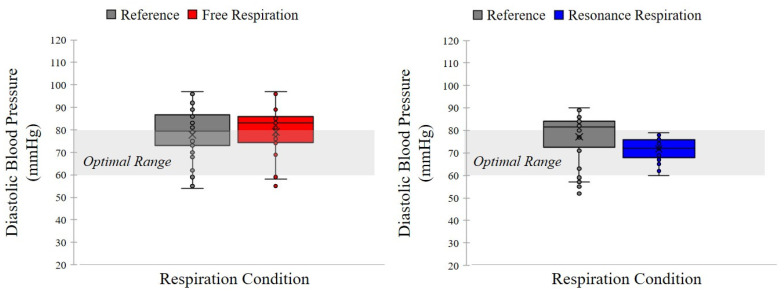
Comparison of results of diastolic blood pressure between free respiration and resonance frequency. The resonance frequency, to diastolic blood pressure, was within the optimal range after respiration task and unrelated to the state before task. However, the free respiration condition was not within the optimal range.

**Figure 8 ijerph-19-15676-f008:**
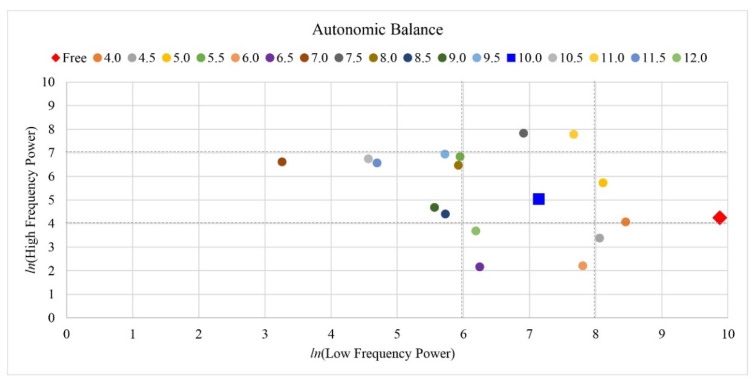
Example of change in autonomic balance in participant 3 causing respiration conditions. The resonance frequency was respiration condition 10, and it was only located in the reference zone. The other respiration condition including free respiration condition was not located in the reference zone.

**Figure 9 ijerph-19-15676-f009:**
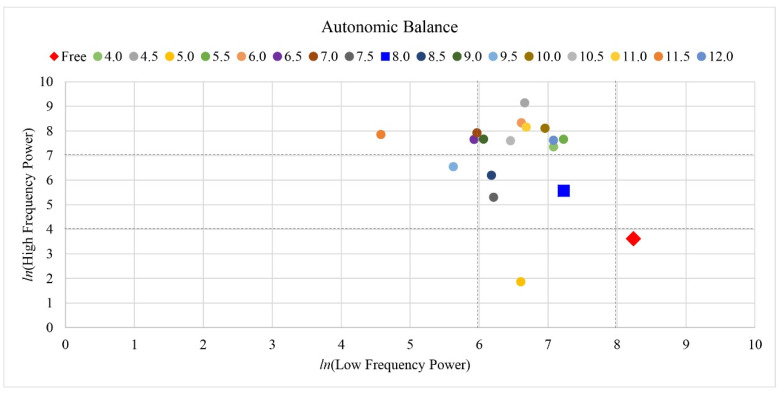
Example of change in autonomic balance in participant 8 causing respiration conditions. The resonance frequency was respiration condition 8, and the other respiration conditions (7.5 and 8.5) with resonance frequency were located in the reference zone.

**Figure 10 ijerph-19-15676-f010:**
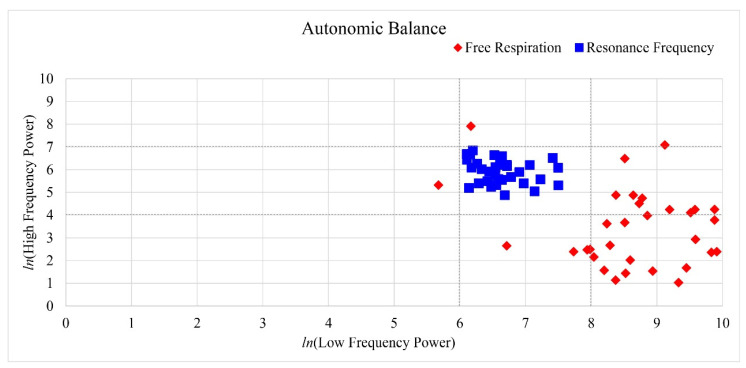
Comparison of results of autonomic balance between free respiration and resonance frequency. The resonance frequency was within the reference zone after respiration task. However, the free respiration condition was not within the reference zone, and almost all participants were located in zones 4 and 5, such as unstable state of autonomic balance (high sympathetic, normal, and low parasympathetic). The time domain LF and HF indices can be quantified during 2 min to 24 h monitoring periods. We obtained the minimum range of ECG data to investigate the short-term harmonic respiration effects; therefore, how changes in autonomic balance evolve over time is not provided here.

**Figure 11 ijerph-19-15676-f011:**
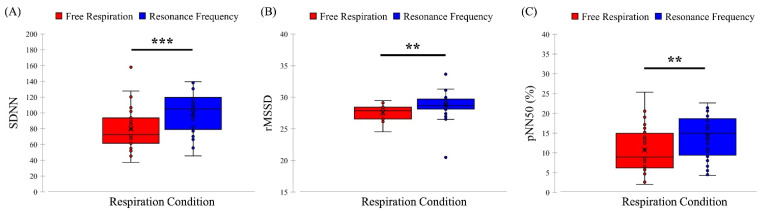
Comparison results of time domain indices of SDNN (**A**), rMSSD (**B**), and pNN50 (**C**) for the free respiration and resonance frequency condition with paired *t*-tests (*** *p* < 0.001, ** *p* < 0.007 adjusted by Bonferroni correction).

**Figure 12 ijerph-19-15676-f012:**
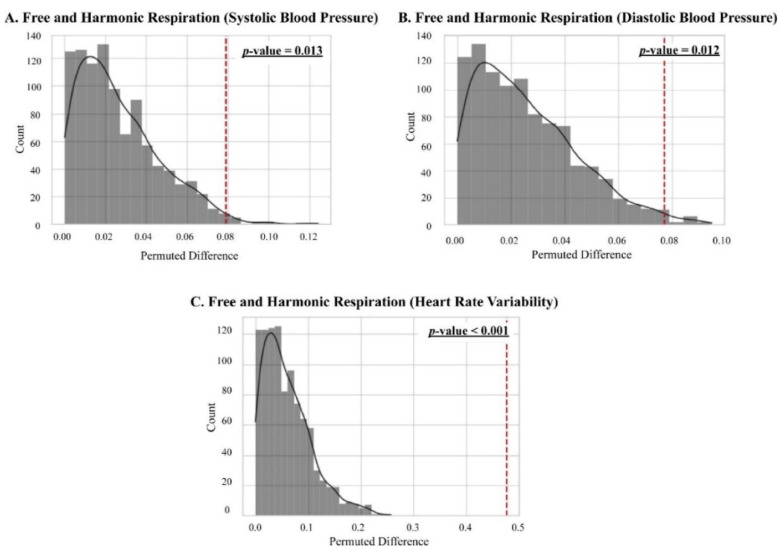
Results of permutation tests for free and harmonic respiration. The underline in the upper right corner of each figure represents empirical *p*-values.

**Figure 13 ijerph-19-15676-f013:**
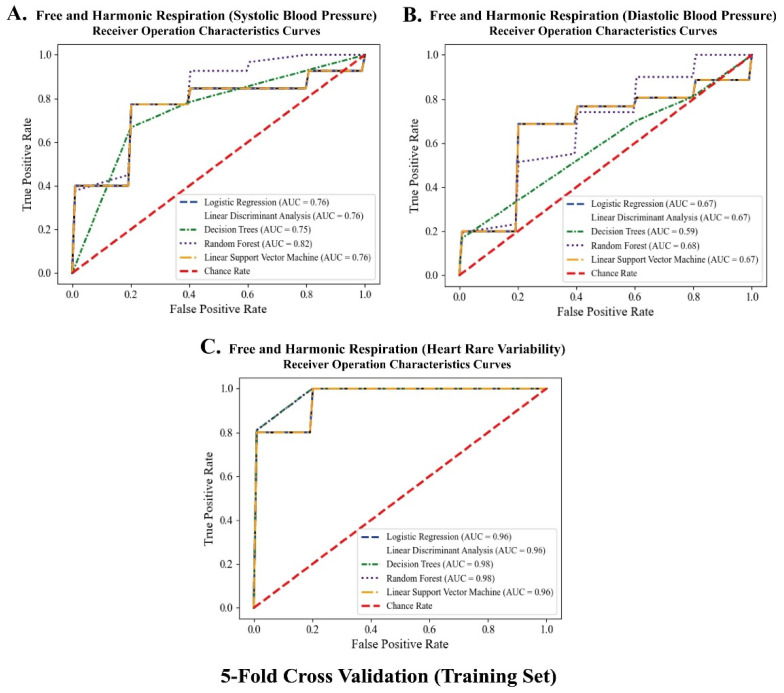
Receiver operating characteristics curves for training dataset with 5-fold cross validation according to five classifiers.

**Figure 14 ijerph-19-15676-f014:**
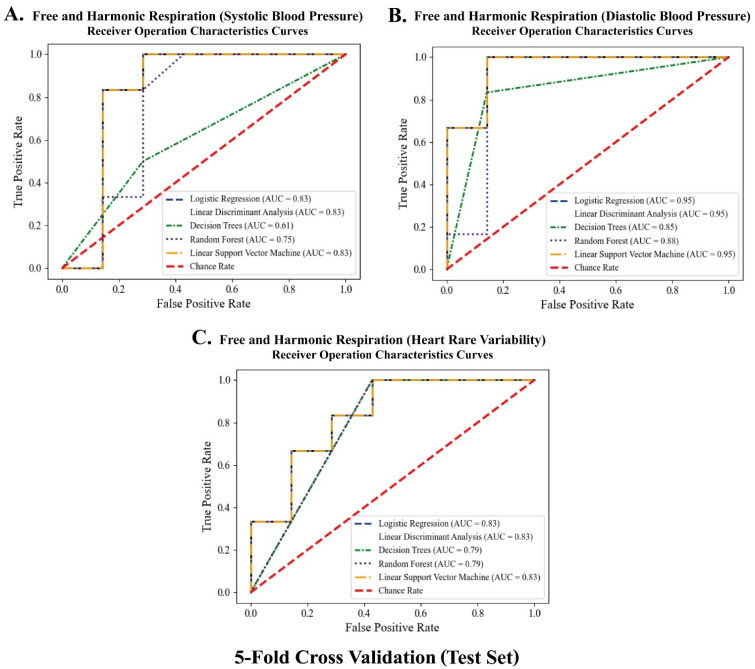
Receiver operating characteristics curves for test dataset with 5-fold cross validation according to five classifiers.

**Figure 15 ijerph-19-15676-f015:**
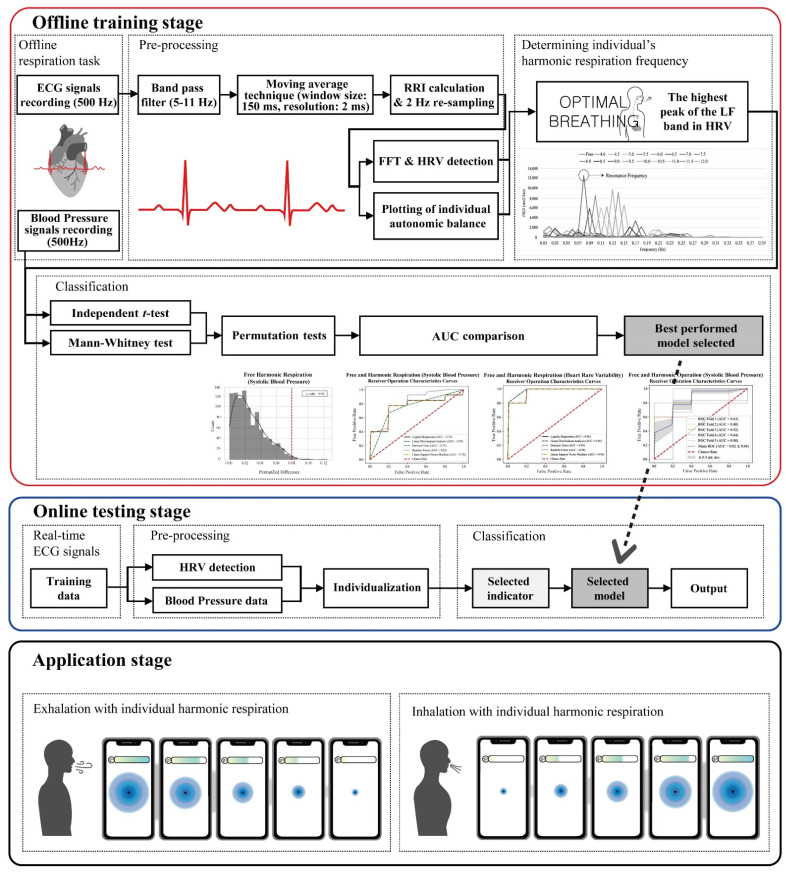
Respiration training system with individual harmonic frequencies in real time.

**Table 1 ijerph-19-15676-t001:** Results of breathing cycle (harmonic frequency) of each participant based on highest HRV spectrum value in LF range (0.04–0.15Hz).

**Participants**	P1	P2	P3	P4	P5	P6	P7	P8	P9	P10	P11
**Breathing Cycle**	5.5	6	10	8	11.5	4	12	8	12	11	10.5
**Participants**	P12	P13	P14	P15	P16	P17	P18	P19	P20	P21	P22
**Breathing Cycle**	7.5	9	6.5	11.5	6	4.5	9	7	11.5	6.5	5
**Participants**	P23	P24	P25	P26	P27	P28	P29	P30	P31	P32	
**Breathing Cycle**	5.5	7	11	7.5	9	6.5	5.5	4.5	5	6.5	

## Data Availability

Data are available on request due to privacy/ethical restrictions.

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
