# Peer review of "Effects of Temporary Respiration Exercise with Individual Harmonic Frequency on Blood Pressure and Autonomic Balance"

_ijerph, 2022, doi:10.3390/ijerph192315676_

Round 1
Reviewer 1 Report
In "Effects of temporary respiration exercise with individual harmonic frequency on blood pressure and autonomic balance," the authors' stated goal is to determine the effect of individual harmonic breathing frequency on blood pressure and the autonomic nervous system. To test evaluate this, the authors used a protocol in which subjects were asked to either freely breathe or breath at 4 to 12 breaths per minute (randomly selected). Based on the data they obtained, they came to two conclusions: (1) blood pressure was stabilized to an optimal range based on the inherent sympathetic rhythm they quantified and (2) autonomic balance was stabilized to a "reference zone" with brief harmonic frequency respiration training (their breathing entrainment protocol).
While the study is very interesting and the data compelling, the manuscript is very difficult to read and understand. I would suggest that the authors find a native English speaker to help them improve the grammar and structure of the manuscript.
Additionally, suggesting that autonomic balance is stabilized by harmonic breathing is significant but, based on the protocol in the manuscript, may not have any significant long-term impact. Obviously, the next study would be to follow these subjects over time and assess stress responses, resilience, and general health. Understanding whether autonomic balance is stabilized with just spectral measures of HRV as the major outcome metric is, perhaps, a grand statement from the data presented. Why were other measures of HRV (RMSSD,SDNN, pNN50, etc.) that are NOT dependent upon spectral measures excluded from analysis? The authors used Python and there is a module called pyHRV (https://pypi.org/project/pyhrv/) that would allow rapid calculation of many different, accepted, algorithms used to quantify HRV. I would hope to see a few different, accepted metrics of HRV quantified and compared before making such a bold assertion.
The authors make the very bold assertion that the mechanism by which stabilization occurs is via changes in heart rate and stroke volume based on a prior study. However, without assessing these changes in their particular protocol, this is merely supposition. Autonomic changes are widespread and include things like: changes in peripheral vascular resistance, altered airway calibre, changes in blood flow to other viscera, etc. All of these may have an impact on "balancing" automatic tone as well but they are not considered in the Discussion. Autonomic control is far more wide-ranging than just impacts on heart function, which is difficult to assess from just the electrocardiogram. Sinus rhythm is certainly a way to integrate cardiorespiratory afferent/efferent flow but may not be the exclusive source of "balance" here. The authors mention stability of blood pressures but I see no explicit definition of stability laid out here (how much variability over time, for example).
Minor concerns:
1. Figure 2 is labeled as "Figure 1", so there are currently two Figure 1's.
2. 4 to 12 breaths per minute seems very low (normal breathing rate in adults is approximately 12 to 20 breaths per minute). Why did the authors use such low breathing rates as part of their protocol? The harmonic relationship of breathing to heart-rate could have been extended into the normal breathing frequency range.
3. Bar graphs are probably not the best way to represent the data in Figures 5, 6, 8, and 9 (where is Figure 7?). I would suggest scatter plots for Figures 5 and 6, and summary box-whisker plots for Figures 8 and 9. We do not really need to see the individual changes in each subject. The optimal ranges can be represented by shading or different colors throughout.
4. Figures 10 and 11 would probably be better represented as a cyclogram (over time) so that your reader can see how the changes in autonomic balance evolve over time.
5. Which Python libraries were used for the machine learning components of the study? Please post any code (and training data) associated with the manuscript as supplemental material. Also, I would suggest making code available on GitHub (https://github.com) if at all possible.
6. Many of the figures are low resolution and somewhat difficult to read (particularly the text in those figures). Please make sure to include high-resolution images for all figures.
7. I prefer not to use "prove" in empirical studies. We test hypotheses using data with the idea that we will DISPROVE a hypothesis (far easier than PROVING a hypothesis) so I typically use the terminology, "Our data lends support to the idea that breathing synchrony with the cardiovascular harmonic frequency can improve overall sympathetic/parasympathetic balance."
Author Response
REVIEW RESPONSE LETTER [International Journal of Environmental Research and Public Health]
Dear Reviewers,
Thank you for reading and reviewing our manuscript, which will help us improve it to a better scientific level. We have made our best efforts to address all the issues pointed out by the reviewers and revised our manuscript as you suggested. Detailed responses to the comments and suggestions are provided below. Once again, we would like to express our gratitude. All revised or added sentences in the manuscript are highlighted in blue font.
Reviewer #1
In "Effects of temporary respiration exercise with individual harmonic frequency on blood pressure and autonomic balance," the authors' stated goal is to determine the effect of individual harmonic breathing frequency on blood pressure and the autonomic nervous system. To test evaluate this, the authors used a protocol in which subjects were asked to either freely breathe or breath at 4 to 12 breaths per minute (randomly selected). Based on the data they obtained, they came to two conclusions: (1) blood pressure was stabilized to an optimal range based on the inherent sympathetic rhythm they quantified and (2) autonomic balance was stabilized to a "reference zone" with brief harmonic frequency respiration training (their breathing entrainment protocol).
While the study is very interesting and the data compelling, the manuscript is very difficult to read and understand. I would suggest that the authors find a native English speaker to help them improve the grammar and structure of the manuscript.
Response: Thank you for reviewing our manuscript and improving our research to a better scientific level. As you suggested, we edited all parts of our manuscript through an English Editing Service with a native English speaker while focusing on improving the grammar and structure of the manuscript.
Q1> Additionally, suggesting that autonomic balance is stabilized by harmonic breathing is significant but, based on the protocol in the manuscript, may not have any significant long-term impact. Obviously, the next study would be to follow these subjects over time and assess stress responses, resilience, and general health.
Response: Thank you for your valuable comments. We agree with the reviewer on the fact that autonomic balance stabilized by the temporary harmonic breathing may not have any significant long-term impact. Thus, we included the following sentence to clarify it in the discussion section.
Page 17 / Line 448
However, our study has the following limitations that need further experiments. Although autonomic balance was stabilized after the temporary harmonic breathing as a function of each individual resonance frequency, long-term effects of the breathing protocol remain to be seen with the further studies. We encourage researchers to investigate the effects of the harmonic breathing on stabilization in autonomic balance with a properly designed experimental protocol for the long-term effects.
Q2> Understanding whether autonomic balance is stabilized with just spectral measures of HRV as the major outcome metric is, perhaps, a grand statement from the data presented. Why were other measures of HRV (RMSSD,SDNN, pNN50, etc.) that are NOT dependent upon spectral measures excluded from analysis?
Response: Thank you for your valuable suggestions. We agree with the reviewer’s opinion and added the protocol and results of the temporal measures of HRV (i.e., RMSSD, SDNN, and pNN50) in 2.3. Data Collection and Signal Processing and 3.4. Time Domain Index of Cardiac Activity sections as follows.
Page 5 / Line 150
The R-peak to R-peak interval (RRI) was calculated by using the difference between R-peak to R-peak, and detected RRIs were filtered as normal to normal (NN) intervals of 600-1200 ms based on the previous work [35]. Time domain indices of cardiac activity such as standard deviation of normal to normal intervals (SDNN), root-mean-square of successive differences (rMSSD), and proportion of NN50 (pNN50) were extracted as follows. (1) The SDNN was computed as the standard deviation across normal RRIs. (2) The rMSSD was calculated as the root mean square of successive differences between normal heartbeats. (3) The pNN50 was calculated by the percentage of adjacent RR intervals that are > 50 ms.
Reference
Kim, J.; Cho, K.; Kim, Y.-K.; Lim, K.-S.; Shin, S.U. Study on Peak Misdetection Recovery of Key Exchange Protocol Using Heartbeat. The Journal of Supercomputing. 2019, 75, 3288–3301. https://doi.org/10.1007/s11227-018-2616-y
Page 12 / Line 334
3.4. Time Domain Parameters of Cardiac Activity
As for the time domain data of cardiac activity, normality assumption was tenable (p > 0.05) and paired t-tests were conducted. The paired t-tests showed that the time domain parameters of cardiac activity in the resonance frequency condition were significantly higher than those in the free respiration condition: SDNN (t[31] = -4.154, p < 0.001, Cohen’s d = 0.780), rMSSD (t[31] = -3.113, p = 0.004, Cohen’s d = 0.700), and pNN50 (t[31] = -3.445, p = 0.001, Cohen’s d = 0.645). The mean (M) and standard deviation (SD) values were as follows: SDNN (Free: M = 79.874, SD = 64.607; Resonance: M = 99.671, SD = 25.330), rMSSD (Free: M = 27.573, SD = 1.169; Resonance: M = 28.737, SD = 2.040), and pNN50 (Free: M = 10.723, SD = 5.526; Resonance: M = 14.191, SD = 5.226), as shown in Figure 11.
As for the Figure 11, please see the attached file.
Q3> The authors used Python and there is a module called pyHRV (https://pypi.org/project/pyhrv/) that would allow rapid calculation of many different, accepted, algorithms used to quantify HRV. I would hope to see a few different, accepted metrics of HRV quantified and compared before making such a bold assertion.
Response: Thank you for your valuable comments. As you suggested, we analyzed some of our data using pyHRV. We found almost similar results between the measures by Labview functions we used and those by the pyHRV modules as follows. We found that the same Welch’s method in calculating power spectrum density both in the Labview and pyHRV module. Please see the difference in results between the two modules in the attached file.
However, it is almost impossible to conduct near-total analytics on various parameters and make new figures within the limited time this journal allows us to revise the manuscript. We strongly believe that core functions in HRV analysis the Labview provides to researchers are very reliable and valid. Thus, we mentioned the following sentence in our manuscript.
Page 5 / Line 163
All signal processing and analysis on HRV parameters were conducted by using the LabVIEW 2010 software (National Instruments, Austin, TX, USA). The Welch’s method was used to calculate power spectrum density in the frequency domain parameters of cardiac activity.
Q4> The authors make the very bold assertion that the mechanism by which stabilization occurs is via changes in heart rate and stroke volume based on a prior study. However, without assessing these changes in their particular protocol, this is merely supposition. Autonomic changes are widespread and include things like: changes in peripheral vascular resistance, altered airway calibre, changes in blood flow to other viscera, etc. All of these may have an impact on "balancing" automatic tone as well but they are not considered in the Discussion. Autonomic control is far more wide-ranging than just impacts on heart function, which is difficult to assess from just the electrocardiogram. Sinus rhythm is certainly a way to integrate cardiorespiratory afferent/efferent flow but may not be the exclusive source of "balance" here. The authors mention stability of blood pressures but I see no explicit definition of stability laid out here (how much variability over time, for example).
Response: We thank reviewer for noticing the important aspects in interpreting our findings. As suggested, we added the following sentences to discuss the issue and mention some limitations in our study as follows.
Page 17 / Line 448
However, our study has the following limitations that need further experiments. Although autonomic balance was stabilized after the temporary harmonic breathing as a function of each individual resonance frequency, long-term effects of the breathing protocol remain to be seen with the further studies. We encourage researchers to investigate the effects of the harmonic breathing on stabilization in autonomic balance with a properly designed experimental protocol for the long-term effects. On top of that, the autonomic balance may be influenced by other potential factors (e.g., changes in peripheral vascular resistance, altered airway calibre, and changes in blood flow to other viscera). It should be noted that potential effects of cardiac vagal activity and vagal nerve mechanisms underlined in regulating cardiac functioning, which are correlated with the autonomic balance, on sympathetic and parasympathetic regulation, are too complex to be understood and have yet to be determined. Thus, future studies are needed to determine that the other factors may be involved in the autonomic balance control with larger number of participants, varied time, repetitions of the respiration training characterized by each individual harmonic frequency, and other measurements of neurotransmitter levels.
In addition, stability of blood pressures means that blood pressures were changed within an optimal range after the individual harmonic breathing. To clarify the definition of stability, we added the following sentence.
Page 9 / Line 275
As shown in Figures 6 and 7, blood pressure was stabilized to the optimal range regardless of whether it was lower or higher than the optimal range. If the blood pressure was high, it decreased to the optimal range and vice versa.
Q5> Figure 2 is labeled as "Figure 1", so there are currently two Figure 1's.
Response: Thank you for your comment. We revised the mistakes and ordering of fifteen figures is correct.
Q6> 4 to 12 breaths per minute seems very low (normal breathing rate in adults is approximately 12 to 20 breaths per minute). Why did the authors use such low breathing rates as part of their protocol? The harmonic relationship of breathing to heart-rate could have been extended into the normal breathing frequency range.
Response: Thank you for this comment. We have mainly focused on potential effects of short-term slow breathing training as a function of each individual harmonic frequency on physiological parameters related to cardiac functions. Then, we investigated the subtle changes in time and frequency domain parameters of cardiac activity and compared them between the free and harmonic respiration frequency condition. Our findings may be a means of optimizing physiological parameters that appear to be associated with autonomic balance. In addition, we agree with your comment that the harmonic relationship of breathing to heart-rate could have been extended into the normal breathing frequency range. To clarify this issue, we added the following sentence in the discussion section.
Page 17 / Line 432
Although, we have mainly focused on potential effects of short-term slow breathing training based on individual harmonic frequencies on autonomic balance, there might have been potentials for extending the controlled harmonic breathing rates from the current slow breathing to normal breathing rates. This is another research topic that needs further experiments and studies to investigate physiological mechanisms underlying the harmonic respiration.
Q7> Bar graphs are probably not the best way to represent the data in Figures 5, 6, 8, and 9 (where is Figure 7?). I would suggest scatter plots for Figures 5 and 6, and summary box-whisker plots for Figures 8 and 9. We do not really need to see the individual changes in each subject. The optimal ranges can be represented by shading or different colors throughout.
Response: Thank you for your kind suggestion. Here, we provide box-whisker plots for the Figures 8 and 9 while we deleted Figure 5 and 6 representing individual changes. Although we made Figure 5 and 6 of the individual changes as scatter plots, the Figure was too complex to read the subtle changes. We considered your comment that “we do not really need to see the individual changes” and deleted the figure that can confuse understanding of potential readers. In addition, we represented the optimal ranges as your suggestion. As for the Figure 6 and 7, please see the attached file.
Page 10 / Line 287
Figure 6. Comparison of results of systolic blood pressure between free respiration and resonance frequency. The resonance frequency was within the optimal range after respiration task, unrelated to the state before the task. On the other hand, the free respiration condition was within the (pre)hypertension state, and thus, was not in the optimal range.
Figure 7. Comparison of results of diastolic blood pressure between free respiration and resonance frequency. The resonance frequency, to diastolic blood pressure, was within the optimal range after respiration task and unrelated to the state before task. However, the free respiration condition was not within the optimal range.
Q8> Figures 10 and 11 would probably be better represented as a cyclogram (over time) so that your reader can see how the changes in autonomic balance evolve over time.
Response: Thank you for your valuable comment. The main purpose of our study was to investigate a feasibility that the short-term respiration training with an individual harmonic frequency can have a positive influence on autonomic balance. Thus, we requested participants to breathe according to our respiration protocols, which led to a small number of NNIs we could optimally calculate the LF and HF parameters in the frequency domain. If we represent the subtle changes in autonomic balance characterized as the ln LF and HF power over time, a reliable resolution in the LF and HF frequency domain cannot be obtained, which may confuse the potential readers and bias the results. To clarify this issue, we added the following sentence in the Figure 10’s caption.
Page 12 / Line 331
Figure 10. Comparison of results of autonomic balance between free respiration and resonance frequency. The resonance frequency was within the reference zone after respiration task. However, the free respiration condition was not within the reference zone, and almost all participants were located in zones 4 and 5, such as unstable state of autonomic balance (high sympathetic, normal, and low parasympathetic). The time-domain LF and HF indices can be quantified during 2 min to 24 h monitoring periods. Since we obtained the minimum range of ECG data to investigate the short-term harmonic respiration effects, how changes in autonomic balance evolve over time is not provided here.
Q9> Which Python libraries were used for the machine learning components of the study? Please post any code (and training data) associated with the manuscript as supplemental material. Also, I would suggest making code available on GitHub (https://github.com) if at all possible.
Response: Thank you for your kind suggestion. Sklearn, pandas, numpy, scipy, and etc. were used for our classification as shown in the following codes we used. We agree with your opinion, but it is difficult to publicly post the algorithm on GitHub. We conducted this research as a consignment research institute from the host research institute with national research funds. We need to get approval from the host research institute, but their research project has been on going and some patent issues still exist. To put it simply, due to the legal, ethical or privacy issues, we have a difficulty in getting approval from the host research institute. Taken together, we stated the following sentence in Data Availability Statement according to MDPI Research Data Policies.
Page 19 / Line 515
Data are available on request due to privacy/ethical restrictions.
Please consider our situation. We will be pleased to provide any codes and data upon request to encourage further studies based on our findings.
Attached are the full codes used in the classification. (Please see the attached file.)
Q10> Many of the figures are low resolution and somewhat difficult to read (particularly the text in those figures). Please make sure to include high-resolution images for all figures.
Response: Thank you for your comment. As suggested, we included high-resolution images with 300 dpi. We suppose that the previous low-resolution was derived from the unexpected conflicts to the submission system. We also uploaded the high-resolution images as separate image files in the submission system.
Q11> I prefer not to use "prove" in empirical studies. We test hypotheses using data with the idea that we will DISPROVE a hypothesis (far easier than PROVING a hypothesis) so I typically use the terminology, "Our data lends support to the idea that breathing synchrony with the cardiovascular harmonic frequency can improve overall sympathetic/parasympathetic balance."
Response: Thank you for your valuable comment. As suggested, we revised the sentences pertinent to our findings as follows.
Page 1 / Line 23
Before:
Our findings demonstrated that blood pressure and autonomic balance were stabilized by temporary harmonic frequency respiration.
After:
Our findings lend support that blood pressure and autonomic balance was improved by temporary harmonic frequency respiration.
Page 18 / Line 478
Before:
Hence, this study proved that people have unique harmonic frequencies that are not limited to a single value.
After:
Our data lend support to the idea that people have unique harmonic frequencies that are not limited to a single value, and breathing synchrony with the cardiovascular harmonic frequency can improve overall sympathetic and parasympathetic balance.

Reviewer 2 Report
This paper investigated effects of modulated respiration on blood pressure and autonomic balance. It was proved that autonomic balance, systolic and diastolic blood pressures can be stabilized with brief respiration training according to harmonic frequency by experiment. The self-regulated respiration system that can control and help stabilize blood pressure and autonomic balance was provided in the study.
This paper has theoretical value and application value. The paper proved the feasibility of stabilizing blood pressure and autonomic balance by harmonic frequency of respiration, and it is useful in the research of reducing mental stress and enhancing human task performance in various fields.
The cited references are mostly relevant but should be more recent. The number of self-citations is suitable. The manuscript’s results are reproducible based on the details given in the methods section. The conclusions are consistent with the evidence and arguments presented. Most of the figures/tables/images/schemes are appropriate and easy to interpret and understand except some figures are too vague.
However, there are still some problems in the paper.
1.The cited references are mostly relevant but should be more recent.
2. The work about respiration training system with individual harmonic frequency can write in more detail.
3.There is no Figure 7 and the words in the Figure 14 and Figure 15 are too vague. These figures should be clearer.
4. In Figure 2,there should be In LF, but what you write is In VLF.
Author Response
REVIEW RESPONSE LETTER [International Journal of Environmental Research and Public Health]
Dear Reviewers,
Thank you for reading and reviewing our manuscript, which will help us improve it to a better scientific level. We have made our best efforts to address all the issues pointed out by the reviewers and revised our manuscript as you suggested. Detailed responses to the comments and suggestions are provided below. Once again, we would like to express our gratitude. All revised or added sentences in the manuscript are highlighted in blue font.
Reviewer #2
This paper investigated effects of modulated respiration on blood pressure and autonomic balance. It was proved that autonomic balance, systolic and diastolic blood pressures can be stabilized with brief respiration training according to harmonic frequency by experiment. The self-regulated respiration system that can control and help stabilize blood pressure and autonomic balance was provided in the study.
This paper has theoretical value and application value. The paper proved the feasibility of stabilizing blood pressure and autonomic balance by harmonic frequency of respiration, and it is useful in the research of reducing mental stress and enhancing human task performance in various fields.
The cited references are mostly relevant but should be more recent. The number of self-citations is suitable. The manuscript’s results are reproducible based on the details given in the methods section. The conclusions are consistent with the evidence and arguments presented. Most of the figures/tables/images/schemes are appropriate and easy to interpret and understand except some figures are too vague. However, there are still some problems in the paper.
Response: Thank you for reviewing our manuscript and improving our research to a better scientific level. As you suggested, we added the references with recently published research articles and improved the image resolution of the figures. We addressed other issues raised and revised them with your comment and suggestion. Detailed responses to the comments and suggestions are provided below. Once again, we would like to express our gratitude. All revised or added sentences in the manuscript are highlighted in blue font.
Q1> The cited references are mostly relevant but should be more recent.
Response: Thank you for this comment. Accordingly, we have added references recently published in the introduction and discussion section. Below are the added references.
Page 19 / Line 518
References
- Chen, X.; Pan, Z. A Convenient and Low-Cost Model of Depression Screening and Early Warning Based on Voice Data Using for Public Mental Health. International Journal of Environmental Research and Public Health. 2021, 18, 6441. https://doi.org/10.3390/ijerph18126441
- Stupak, R.; Dobroczyn´ski, B. From Mental Health Industry to Humane Care. Suggestions for an Alternative Systemic Approach to Distress. International Journal of Environmental Research and Public Health. 2021, 18, 6625. https://doi.org/10.3390/ijerph18126625
- Santos, E.G.d.O.; Queiroz, P.R.; Nunes, A.D.d.S.; Vedana, K.G.G.; Barbosa, I.R. Factors Associated with Suicidal Behavior in Farmers: A Systematic Review. International Journal of Environmental Research and Public Health. 2021, 18, 6522. https://doi.org/10.3390/ijerph18126522
- Gholamrezaei, A.; Diest, I.V.; Aziz, Q.; Pauwels, A.; Tack, J.; Vlaeyen, J. W.; Oudenhove, L.V. Effect of slow, deep breathing on visceral pain perception and its underlying psychophysiological mechanisms. Neurogastroenterology & Motility, 2022, 34, e14242. https://doi.org/10.1111/nmo.14242
- Juventin, M.; Ghibaudo, V.; Granget, J.; Amat, C.; Courtiol, E.; Buonviso, N. Respiratory influence on brain dynamics: the preponderant role of the nasal pathway and deep slow regime. Pfügers Archiv - European Journal of Physiology, 2022, 1-13. https://doi.org/10.1007/s00424-022-02722-7
- Laborde, S.; Allen, M.S.; Borges, U.; Dosseville, F.; Hosang, T.J.; Iskra, M.; Mosley, E.; Salvotti, C.; Spolverato, L.; Zammit, N.; Javelle, F. Effects of voluntary slow breathing on heart rate and heart rate variability: A systematic review and a meta-analysis. Neuroscience & Biobehavioral Reviews. 2022, 138, 104711. https://doi.org/10.1016/j.neubiorev.2022.104711
- Laborde, S.; Allen, M.S.; Borges, U.; Hosang, T.J.; Furley, P.; Mosley, E.; Dosseville, F. The Influence of Slow-Paced Breathing on Executive Function. Journal of Psychophysiology. 2022, 36, 1-13. https://doi.org/10.1027/0269-8803/a000279
- Sevoz-Couche, C.; Laborde, S. Heart rate variability and slow-paced breathing: when coherence meets resonance. Neuroscience & Biobehavioral Reviews, 2022, 135, 104576. https://doi.org/10.1016/j.neubiorev.2022.104576
- Szulczewski, M.T. Transcutaneous Auricular Vagus Nerve Stimulation Combined With Slow Breathing: Speculations on Potential Applications and Technical Considerations. Neuromodulation: Technology at the Neural Interface. 2022, 25, 380-394 https://doi.org/10.1111/ner.13458
- You, M.; Laborde, S.; Zammit, N.; Iskra, M.; Borges, U.; Dosseville, F.; Vaughan, R.S. Emotional Intelligence Training: Influence of a Brief Slow-Paced Breathing Exercise on Psychophysiological Variables Linked to Emotion Regulation. International Journal of Environmental Research and Public Health. 2021, 18, 6630.https://doi.org/10.3390/ijerph18126630
Q2> The work about respiration training system with individual harmonic frequency can write in more detail.
Response: Thank you for your valuable comment. As suggested, we have added how the respiration training system with individual harmonic frequency works in detail as follows.
Page 15 / Line 383
A respiration training system with individual harmonic frequencies was developed using Visual C++ 2013 and OpenCV 3.0.0, as shown in Figure 15. During the offline training stage, users are asked to perform slow breathing training according to the proposed slow breathing frequencies while their blood pressure and ECG data are recorded through wearable sensors. Resampling-based permutation tests validate whether reliable significant changes are observed among the breathing conditions. Then, an individual cardiovascular harmonic frequency was determined and only significant parameters are trained and classified. The system automatically selects the best-performed machine learning algorithm. During the online testing stage, users are requested to carry out only two free and slow breathing as a function of the determined harmonic frequency for about 7 min with an interval of 1 min. The classifier selected from the best-performed model automatically categorizes significant parameters from blood pressure and ECG data into a binary class between the free and individual harmonic breathing. Then, classification results obtained from the offline and online stage are compared. Unless a clear classification performance is not reported or if less than mean AUC of 0.70 is reported, the classifier is initialized and a fine-tuning stage of individualization begins to optimize the parameters with different weights. The mobile healthcare application could determine the individual resonance frequency, which contributes to stabilizing the autonomic balance based on the classification procedures.
Q3> There is no Figure 7 and the words in the Figure 14 and Figure 15 are too vague. These figures should be clearer.
Response: Thank you for this comment. As suggested, we included high-resolution images with 300 dpi. We suppose that the previous low-resolution was derived from the unexpected conflicts to the submission system. We also uploaded the high-resolution images as separate image files in the submission system. As for the Figure 14, we divided one figure into two parts to improve readability as well as to create high-resolution images.
Page 14 / Line 377
Q4> In Figure 2,there should be In LF, but what you write is In VLF.
Response: We thank reviewer for noticing the important part in the Figure was written by mistake. We revised the part from ln VLF to ln LF as follows. The order of the Figure was assigned as Figure 3 since we rearranged the order of all Figures. As for the Figure 3, please see the attached file.
Page 6 / Line 168

Round 2
Reviewer 1 Report
Thank you for your revisions to the manuscript as it is now greatly improved. My only remaining concern is that the legends for the graphs in Figures 13 and 14 obscure the plots. Please move them over a bit more so that all the graphs can be clearly seen. Also, I would appreciate it if the authors made their Python data analysis script/notebook available as supplementary material. This should not have any PHI from patients so I do not understand the concern about confidentiality. Thank you.
Author Response
REVIEW RESPONSE LETTER [International Journal of Environmental Research and Public Health]
Dear Reviewers,
Thank you for reading and reviewing our manuscript, which will help us improve it to a better scientific level. We have made our best efforts to address all the issues pointed out by the reviewers and revised our manuscript as you suggested. In addition, we carried out the spell-check function in MS word and revised our manuscript. Detailed responses to the comments and suggestions are provided below. Once again, we would like to express our gratitude.
Reviewer #1
Thank you for your revisions to the manuscript as it is now greatly improved. My only remaining concern is that the legends for the graphs in Figures 13 and 14 obscure the plots. Please move them over a bit more so that all the graphs can be clearly seen. Also, I would appreciate it if the authors made their Python data analysis script/notebook available as supplementary material. This should not have any PHI from patients so I do not understand the concern about confidentiality. Thank you.
Response: Thank you for reviewing our manuscript and improving our research to a better scientific level. As you suggested, we moved the legends of the graphs in Figure 13 and 14 without obscuring the plots (Please see the attached file). In addition, we provided the Python data analysis script for the binary classification as a supplemental material as suggested. Thank you for your efforts and time in reviewing our paper.
